# On-the-Fly Adaptive Distillation of Transformer to Dual-State Linear Attention for Long-Context LLM Serving

Yeonju Ro [1]   Zhenyu Zhang [1]   Souvik Kundu [2]   Zhangyang Wang [1]   Aditya Akella [1]

## Abstract

Large language models (LLMs) excel at capturing global token dependencies via self-attention but face prohibitive compute and memory costs on lengthy inputs. While sub-quadratic methods (e.g., linear attention) can reduce these costs, they often degrade accuracy due to overemphasizing recent tokens. In this work, we first propose *dual-state linear attention* (**DSLA**), a novel design that maintains two specialized hidden states—one for preserving historical context and one for tracking recency—thereby mitigating the short-range bias typical of linear-attention architectures. To further balance efficiency and accuracy under dynamic workload conditions, we introduce **DSLA-*Serve***, an online *adaptive distillation* framework that progressively replaces Transformer layers with DSLA layers at inference time, guided by a sensitivity-based layer ordering. DSLA-*Serve* uses a chained fine-tuning strategy to ensure that each newly converted DSLA layer remains consistent with previously replaced layers, preserving the overall quality. Extensive evaluations on commonsense reasoning, long-context QA, and text summarization demonstrate that DSLA-*Serve* yields **2.3×** faster inference than Llama2-7B and **3.0×** faster than the hybrid Zamba-7B, while retaining comparable performance across downstream tasks. Our ablation studies show that DSLA's dual states capture both global and local dependencies, addressing the historical-token underrepresentation seen in prior linear attentions. Codes are available at: https://github.com/utnslab/DSLA-Serve.

## 1. Introduction

Large language models (LLMs) have become indispensable for tasks such as advanced reasoning, code generation, and multi-turn dialogue. However, *long-sequence* inference remains a major bottleneck due to the $\mathcal{O}(T^2)$ complexity of self-attention (Vaswani, 2017), which evaluates pairwise relationships among all tokens in a sequence. Alongside heavy compute cost, storing key-value ($KV$) caches for extended contexts forces memory usage to grow linearly with the number of tokens, significantly challenging scalability.

To reduce inference complexity, prior work explores both *system-level* optimizations (e.g., cache management and GPU kernels (Ye et al., 2025; Dao et al., 2022; Kwon et al., 2023b)) and *algorithmic* approaches (Leviathan et al., 2023; Cai et al., 2024a; Kudugunta et al., 2023; Xiao et al., 2023b; Zhang et al., 2023). Among algorithmic solutions, *linear attention* has gained traction (Peng et al., 2023; Wang et al., 2020; Gu & Dao, 2023), maintaining a fixed-size hidden state to update token embeddings in $\mathcal{O}(T)$ time. This makes such models appealing for large-context processing.

Despite their efficiency, purely linear attention models often underperform on tasks requiring careful long-range interactions (e.g., large-scale text retrieval) (Waleffe et al., 2024; Gu & Dao, 2023; Lieber et al., 2024). Their fixed hidden states impose a capacity bottleneck, especially when the task demands rich cross-token relationships. Hybrid architectures have thus emerged, mixing a few self-attention and linear-attention layers (Lieber et al., 2024; Ren et al., 2024; hug). However, these hybrids typically rely on a *static* design, which cannot flexibly trade off compute cost versus accuracy on the fly. Meanwhile, recent attempts to *distill* transformers into linear attention (Bick et al., 2024; Zhang et al., 2024a; Wang et al., 2024a) either incur significant accuracy drops when fully replacing self-attention, or remain constrained by bottleneck layers if only partially converted. Consequently, current approaches do not adapt well to the variable demands of real-world inference pipelines.

A naturally emerging, intuitively promising solution is then to *dynamically* adapt architectures at inference time, converting self-attention layers into linear-attention analogs only under memory or latency pressure. Yet achieving such "on-

---

[1]The University of Texas at Austin [2]Intel Labs. Correspondence to: Yeonju Ro <yro@cs.utexas.edu>.

*Proceedings of the 42nd International Conference on Machine Learning*, Vancouver, Canada. PMLR 267, 2025. Copyright 2025 by the author(s).

the-fly" adaptive distillation is non-trivial. First, we must decide how many layers can be replaced without unacceptably harming accuracy. Second, mismatches between training and inference architectures easily degrade performance if not carefully addressed. Third, since linear-attention methods often exhibit a strong bias toward recent tokens (see §3), naive distillation may fail to capture the global attention patterns that long contexts demand.

**Our Contributions.** In this work, we address these challenges through:

- **Dual-State Linear Attention (DSLA).** We first analyze how single-state linear attention overemphasizes recent tokens and propose a new module with *two* specialized states: one for preserving *historical* context and the other tracking *recency*. Driven by a constrastive regularization, this design alleviates the short-range bias and better emulates full self-attention.

- **Adaptive Distillation via DSLA-*Serve*.** We then introduce DSLA-*Serve*, a framework that *adaptively* converts Transformer layers into DSLA at inference time, guided by a sensitivity- based layer ordering. This "on-the-fly" approach allows different workloads or prompt lengths to flexibly trade off accuracy and efficiency.

- **Chained Fine-Tuning for Consistency.** To minimize performance drops during layer conversion, DSLA-*Serve* employs a *progressive* distillation procedure. Layers are distilled in precisely the order they will be converted at inference, ensuring each newly replaced layer is compatible with previously converted ones.

- **Extensive Evaluations.** Across multiple datasets, DSLA-*Serve* achieves a $2.29\times$ and $3.0\times$ reduction in end-to-end latency compared to Llama2-7B and Zamba-7B baselines, respectively, while maintaining comparable performance on tasks spanning reasoning, code, and long-context understanding.

## 2. Background

**Notations.** We denote matrices with bold uppercase letters (e.g., $\mathbf{X}$), vectors with bold lowercase letters (e.g., $\mathbf{x}$), and scalars with lowercase letters (e.g., $x$).

**Linear Attention.** A primary driver behind sub-quadratic complexity is *linear attention*, which replaces the $\mathcal{O}(T^2)$ all-pairs interactions of self-attention with recurrences or kernel-based factorizations (Child et al., 2019; Beltagy et al., 2020; Zaheer et al., 2020; Choromanski et al., 2020; Katharopoulos et al., 2020). Recent approaches often adapt *state-space models* (SSMs) (Gu & Dao, 2023; Dao & Gu, 2024) or *RNN gating* (Peng et al., 2023) to maintain a *fixed-size* key-value (KV) state that updates incrementally across tokens. While this design yields $\mathcal{O}(T)$ computation, a single recurrent

state can overemphasize the most recent context, limiting deeper cross-token learning.

**Gated Linear Attention (GLA).** To mitigate underperformance in standard SSMs, GLA (Yang et al., 2023) introduces *data-dependent gating*, adding a learnable forget gate $\mathbf{G}_t \in \mathbb{R}^{d \times d}$ that modulates the previous hidden state $\mathbf{S}_{t-1}$:

$$\mathbf{S}_t = \mathbf{G}_t \odot \mathbf{S}_{t-1} + \mathbf{k}_t^\top \mathbf{v}_t,$$

where $\mathbf{G}_t$ is formed from outer products of vectors $\boldsymbol{\alpha}_t, \boldsymbol{\beta}_t$, and $\mathbf{k}_t, \mathbf{v}_t$ are the key/value embeddings at time $t$. Although this gating can help retain some historical information, GLA still fundamentally relies on *one* hidden state, often losing mid- or far-context representations.

**Knowledge Distillation into Linear Attention.** To narrow performance gaps between quadratic self-attention and linear methods, some works apply distillation (Wang et al., 2024a; Zhang et al., 2024a). However, these static conversions do not address *dynamic* inference scenarios, where the model must flexibly trade accuracy for reduced memory cost under changing request loads or context lengths.

**Hybrid Architectures** Another approach is to *mix* self-attention with linear layers at different layers or heads (Waleffe et al., 2024; Lieber et al., 2024; hug). While hybrid models can mitigate the drawbacks of purely linear or purely quadratic methods, their *fixed* attention layouts remain unsuitable when inference loads fluctuate.

***By contrast***, our work aims to (i) *improve* the typical single-state design of linear attention through a *dual-state* mechanism, thereby alleviating recency bias, and (ii) enable *on-the-fly* conversion of self-attention layers so that the model can dynamically trade off accuracy and efficiency based on system constraints. We demonstrate how single-state designs exhibit short-range bias through an attention analysis, then show how *dual-state* linear attention (DSLA) successfully mitigates this issue. We finally (iii) incorporate an *adaptive* inference framework that progressively replaces self-attention with DSLA layers under high resource load, ensuring scalable and efficient LLM serving without sacrificing crucial long-context performance.

## 3. Motivational Analyses

In this section, we present two key motivating observations:

- how single-state linear attention tends to overemphasize recent tokens at the expense of historical context;

- why LLM serving systems would desire to dynamically adapt to changing resource constraints rather than relying on static architectures.

**Attention Score Recency Analysis.** Gated linear attention (GLA) (Yang et al., 2023) updates the hidden state and

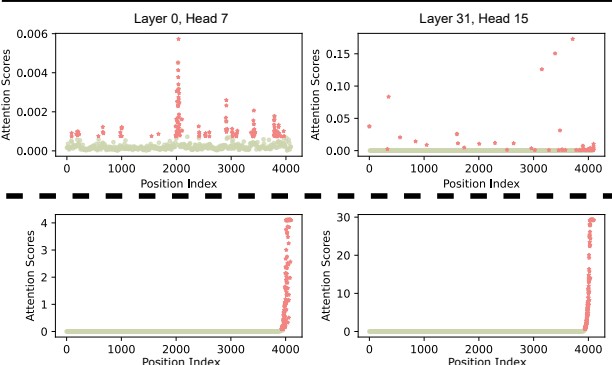

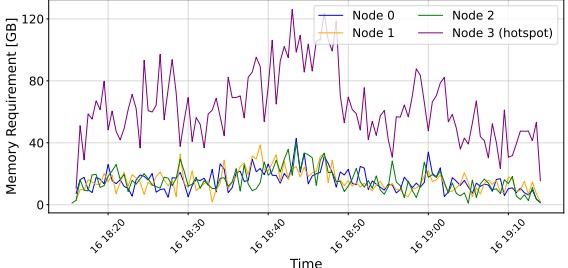

*Figure 2.* Illustration of resource usage spikes (memory) over time and across nodes. Long-lived or bursty sessions can heavily strain certain GPUs.

*Figure 1.* Attention scores for Llama2-7B (top) vs. GLA (bottom). The latter is distilled from Llama2-7B with using 1B tokens and KL divergence w.r.t. attention outputs. We examine several randomly selected attention heads/layers on 16 samples from C4 (sequence length 4096). The horizontal axis is the token index, and the vertical axis shows the attention scores to past tokens from the most recent query. GLA focuses heavily on near tokens, while the Transformer maintains more balanced coverage of earlier positions.

output at time step $t$ via:

$$\mathbf{S}_t = \mathbf{G}_t \odot \mathbf{S}_{t-1} + \mathbf{k}_t^\top \mathbf{v}_t, \tag{1}$$

$$\mathbf{o}_t = \mathbf{q}_t \mathbf{S}_t, \tag{2}$$

where $\mathbf{G}_t$ is a learnable gating term in $\mathbb{R}^{d \times d}$, and $\mathbf{k}_t, \mathbf{v}_t, \mathbf{q}_t$ are the key, value, and query representations. From these, the final query $\mathbf{q}_T$ produces

$$\mathbf{o}_T = \sum_{t=0}^{T} \boldsymbol{\sigma}_t \mathbf{v}_t, \quad \boldsymbol{\sigma}_t = (\mathbf{q}_T \odot \mathbf{k}_t) \Big( \prod_{\tau=t+1}^{T} \mathbf{G}_\tau \Big), \tag{3}$$

illustrating how the gating products affect attention to each past token $v_t$.

Figure 1 compares attention scores for $\mathbf{q}_T$ across multiple heads in Llama2-7B (top) vs. GLA (bottom) on sequences of length 4096. In many Transformer heads, the attention distribution extends across earlier tokens (including middle or far context). By contrast, GLA consistently places disproportionate weight on recent tokens. We hypothesize that this stems from GLA's reliance on a *single* compressive state $\mathbf{S}_t$ with a fixed dimension, which must simultaneously encode all prior context. Over long sequences, the gating $\mathbf{G}_t$ often pushes the model to "forget" older segments (Ben-Kish et al., 2024; Azizi et al., 2025). In next-token prediction settings, focusing on local tokens often suffices for short contexts, but can be detrimental when crucial information resides far back in the input. We notice a concurrent work (Wang et al., 2024b) has drawn similar "recency bias" observations on state-space models via theoretical analysis.

**Need for Adaptive Models.** We also highlight the necessity of *adaptable* inference architectures through a real serving trace (Appendix C), shown in Figure 2. The y-axis indicates resource usage, measured over time (x-axis) across

multiple nodes. In production-scale deployments, load fluctuations arise due to:

- **Temporal variability:** Sporadic bursts of lengthy requests or multiple concurrent sessions can dramatically increase memory usage when each session accumulates a large $KV$ cache.

- **Spatial imbalance:** Session affinity in multi-turn chat systems can repeatedly assign requests from the same user to a single GPU, causing that node to saturate while other nodes remain underutilized.

A common fallback is to over-provision GPU capacity, which is both costly and under-utilized most of the time. Conversely, *fully* replacing self-attention with linear layers reduces memory but can degrade performance on tasks demanding nuanced cross-token relationships. Hence, a flexible mechanism is needed to *partially* convert layers *on the fly*, balancing efficiency and accuracy in response to real-time resource pressure. Existing static architectures are ill-suited for such dynamic scenarios, motivating our framework that progressively substitutes a Transformer's self-attention layers with linear-attention analogs under high load, while retaining more accurate (i.e., quadratic) layers when resources are sufficient.

## 4. Methodology

Motivated by the recency bias in single-state linear attention (§3), this section introduces our proposed *Dual-State Linear Attention* (DSLA), which maintains both "history" and "recent" contexts. We then describe an *adaptive* LLM-serving framework, DSLA-*Serve*, that *progressively* converts self-attention layers into DSLA layers in response to dynamic workload demands. Figure 8 gives a high-level overview.

### 4.1. Dual-State Linear Attention (DSLA)

Gated Linear Attention (GLA) (Yang et al., 2023) uses a single hidden state $\mathbf{S}_t$ updated by a forget gate, which causes earlier tokens to be truncated over long sequences. To mitigate this *recency bias*, we propose **D**ual-**S**tate **L**inear

Attention (**DSLA**), where each layer maintains *two* states:

$$\mathbf{S}_t^1 = \mathbf{G}_t^1 \odot \mathbf{S}_{t-1} + \mathbf{k}_t^\top \mathbf{v}_t, \quad \mathbf{S}_t^2 = \mathbf{G}_t^2 \odot \mathbf{S}_{t-1} + \mathbf{k}_t^\top \mathbf{v}_t. \tag{4}$$

Here, $\mathbf{G}_t^1, \mathbf{G}_t^2 \in \mathbb{R}^{d \times d}$ serve as data-driven forget mechanisms; $\mathbf{k}_t, \mathbf{v}_t$ are the key/value vectors at time $t$. We then blend these two states in the output:

$$\mathbf{o}_t = \mathbf{q}_t \left( \gamma \cdot \mathbf{S}_t^1 + (1 - \gamma) \mathbf{S}_t^2 \right), \tag{5}$$

where $\mathbf{q}_t$ is the query, and $\gamma$ is a *learnable coefficient* that determines the relative weight of each state. By default, we initialize $\mathbf{G}_t^1$ (the "history" gate) to be closer to the identity matrix, helping preserve older context, while $\mathbf{G}_t^2$ (the "recency" gate) is randomly initialized (e.g., from $\mathcal{N}(0, \sigma^2)$) to have a broader forgetting effect. This *dual-state* design offers more capacity to store mid- or far-range tokens, while still benefiting from the linear-time update rule.

Notably, attention patterns vary significantly *across* layers in a deep model (Ren et al., 2024; Waleffe et al., 2024). Early layers often focus on local context, middle layers tend to retrieve from mid-range segments, and later layers aggregate more global representations. To accommodate these differing roles, we introduce a *learnable coefficient $\gamma$ per layer*, which adjusts the relative emphasis on our two states (history vs. recent). Formally, at the final token $T$, the DSLA output is:

$$\mathbf{o}_T = \sum_{t=0}^{T} \left( \gamma \, \boldsymbol{\sigma}_t^1 + (1 - \gamma) \, \boldsymbol{\sigma}_t^2 \right) \mathbf{v}_t, \tag{6}$$

where

$$\boldsymbol{\sigma}_t^1 = (\mathbf{q}_T \odot \mathbf{k}_t) \left( \mathbf{G}_T^1 \odot \mathbf{G}_{T-1}^1 \ldots \odot \mathbf{G}_{t+1}^1 \right), \tag{7}$$

$$\boldsymbol{\sigma}_t^2 = (\mathbf{q}_T \odot \mathbf{k}_t) \left( \mathbf{G}_T^2 \odot \mathbf{G}_{T-1}^2 \ldots \odot \mathbf{G}_{t+1}^2 \right). \tag{8}$$

By learning $\gamma$ alongside the gating parameters, each layer can balance long-range and local information differently. We interpret $\|\gamma \, \boldsymbol{\sigma}_t^1 + (1 - \gamma) \, \boldsymbol{\sigma}_t^2\|_2$ as the *attention score* to token $\mathbf{v}_t$ for the final query $\mathbf{q}_T$, reflecting a superposition of history and recency signals tailored to that layer's role.

## 4.2. Specializing the Two Hidden States

While two hidden states $\mathbf{S}_t^1, \mathbf{S}_t^2$ expand the capacity beyond single-state GLA, we also seek to ensure that each state *specializes* to capture different parts of the context. If $\mathbf{S}_t^1$ and $\mathbf{S}_t^2$ converge to the same gating or focus on overlapping token spans, the dual-state advantage diminishes.

**Why Two States Suffice?**  In principle, multiple additional states could be introduced, each with its own gating matrix. However, in our experiments, we observe that splitting the capacity into two *distinct* states already provides significant

gains: one state tends to preserve *global* or *historical* context, while the other focuses on *recency* or *local* patterns (cf. §5.5). Empirically, using more than two states showed diminishing returns, increasing training overhead and model complexity without sizable accuracy improvements. Once again, **the key is to properly specialize the two states**.

**Contrastive Regularization.**  To drive specialization, we combine a *distillation* term with a *contrastive* penalty. Let $\mathbf{o}_t$ denote our DSLA output at position $t$, and $\mathbf{o}_t^{(\text{gt})}$ be the self-attention output from the teacher Transformer. Define:

$$\mathcal{L}_{dist} = \mathcal{D}_{KL}\left(\mathbf{o}_t \,\|\, \mathbf{o}_t^{(\text{gt})}\right), \quad \mathcal{L}_{cont} = \text{sim}\left(\mathbf{G}_t^1, \mathbf{G}_t^2\right), \tag{9}$$

where $\text{sim}(\cdot, \cdot)$ measures cosine similarity between the two gating matrices $\mathbf{G}_t^1, \mathbf{G}_t^2$ at time $t$. Minimizing $\mathcal{L}_{dist}$ aligns DSLA's outputs with the teacher's, while minimizing $\mathcal{L}_{cont}$ *reduces* the similarity of gating functions across the two states, encouraging them to attend to different aspects of the sequence. The total loss is:

$$\mathcal{L} = \mathcal{L}_{dist} + \lambda \mathcal{L}_{cont}, \tag{10}$$

with a hyperparameter $\lambda$ balancing teacher alignment and state differentiation. During training, the *contrastive* term ensures that the two gates do not collapse to similar solutions, while the *distillation* term aligns the overall DSLA output with the teacher's self-attention. In practice, we observe that DSLA converges in roughly the same number of steps as single-state GLA, indicating minimal overhead.

**Interpretation.**  Qualitative attention-score analyses (Figure 5) show that the two gates yield distinct patterns: $\mathbf{S}_t^1$ retains capacity for earlier tokens, whereas $\mathbf{S}_t^2$ adaptively highlights recent context. Together, they more closely replicate full Transformer attention while retaining the linear-time update rule. We next explain how we use DSLA at inference time in a partial or fully converted manner (§4.3), enabling efficient serving across diverse workloads.

## 4.3. Adaptive Layer-wise Conversion with DSLA-*Serve*

While converting *all* self-attention layers to DSLA maximizes speedups and memory savings, it also carries higher risk of accuracy degradation. In practice, the optimal trade-off between efficiency and accuracy can vary with both workload and application constraints. To handle this in fine granularity, we propose an *automatic* conversion framework, DSLA-*Serve*, that progressively replaces Transformer layers with DSLA at inference time (Figure 8).

**Layer Ranking via Sensitivity.**  Before deployment, we distill each layer from the teacher model (e.g., a Transformer) into a DSLA counterpart, then measure how much

replacing that layer alone impacts validation accuracy (or perplexity). For a more fine-grained indicator, we compute *attention entropy*:

$$\text{Entropy}(\mathbf{A}) = \sum_{i=0}^{T} \mathbf{A}_{T,i} \log \mathbf{A}_{T,i}, \qquad (11)$$

where $\mathbf{A}_{T,i}$ is the attention score matrix for the final query $\mathbf{q}_T$. Lower entropy often indicates a layer focusing on fewer tokens, making it "less sensitive" to linearization. We sort layers in ascending order of sensitivity.

**Adaptive Conversion Logic.** During runtime, DSLA-*Serve* monitors memory load and query lengths. When the system detects high GPU pressure, it begins converting layers from the *least sensitive* end of the queue to DSLA form, thereby reducing memory usage by removing the large key-value ($KV$) cache footprint of those layers. This process continues until either:

1. **Memory Relief Achieved:** Enough layers are converted that GPU usage falls below a threshold; or

2. **Accuracy at Risk:** A certain service-level objective (SLO) on quality (e.g., perplexity or BLEU score) is at risk, so DSLA-*Serve* halts further conversion.

**Prioritizing Long Contexts.** Many real-world requests are short and do not stress memory resources. By contrast, large prompts and multi-turn sessions can rapidly inflate the $KV$ cache. Thus, DSLA-*Serve* focuses *first* on converting layers for those *long* queries, yielding outsized gains in memory and speed. Our experiments (Tables 1-2) show that partial DSLA substitution for long prompts retains strong performance while substantially reducing latency.

**Connection to Real-World Load.** As discussed in §3, production workloads exhibit both temporal and spatial load imbalances. By flexibly converting a growing fraction of layers to DSLA, DSLA-*Serve* can quickly address transient spikes in GPU usage, then stop once the system stabilizes. This "pay as you go" strategy avoids static overprovisioning or fully switching to linear layers (which may excessively degrade accuracy). We next describe how DSLA-*Serve* ensures consistency across partial and fully converted configurations (§4.4), followed by batching considerations (§4.5).

### 4.4. Chained Fine-Tuning for Seamless Conversion

Although adaptively converting layers during inference offers a flexible memory–accuracy trade-off, naively training DSLA replacements for each layer in isolation can lead to inconsistent behavior when multiple layers are simultaneously converted. In other words, replacing layer $i$ while assuming all other layers are Transformers at training time will not match actual scenarios where one or more other

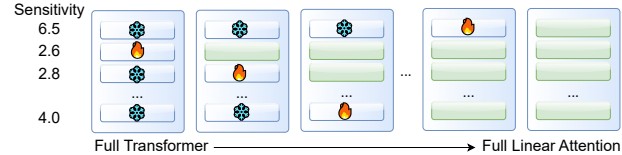

*Figure 3.* **Chained Fine-Tuning.** We replace layers in ascending sensitivity order, freezing all other layers. Upon completion of each stage, we commit the newly trained DSLA layer so that subsequent stages see the updated architecture.

layers have already been converted. To address this, we adopt a *chained* fine-tuning approach that ensures consistency between training and deployment architectures. As illustrated in Figure 3, we process with the following recipe:

1. **Layer Ranking.** Identify and order the layers by their sensitivity (cf. §4.3).

2. **Replace and Finetune.** Convert the first (lowest-sensitivity) layer to DSLA, freeze all other layers, and fine-tune until convergence.

3. **Update Baseline.** Permanently replace that layer with its DSLA counterpart, so subsequent layers see the updated partial-linear architecture.

4. **Iterate.** Move to the next layer in the queue, repeating the same procedure.

By the time we finish training the most sensitive layer, the model can handle any partial combination of DSLA and Transformer layers without unexpected degradations. This yields a *family* of progressively converted models whose partial configurations are consistent at test time.

This approach prevents train–test mismatch that would otherwise arise if multiple DSLA layers are used in deployment but were never jointly seen during training. Empirically, our experiments confirm that chained fine-tuning mitigates accuracy drops and ensures a smooth trade-off between memory usage and performance.

### 4.5. Batched Inference Considerations

Batch decoding (Yu et al., 2022; Kwon et al., 2023a) is a common practice for high-throughput LLM services, where multiple generation *requests* (each corresponding to a different user prompt) are processed in parallel. However, *partial* layer conversion can create branching paths—some requests may need a DSLA layer while others still rely on self-attention if they have not (or do not) trigger conversion. These requests cannot be fused into a single GEMM pass because DSLA parameters differ from those of self-attention.

**Re-Batching Overhead.** When the model encounters a layer where some requests use DSLA while others still use self-attention, DSLA-*Serve* separates them into two sub-batches. After that layer, the sub-batches may be merged

again for subsequent shared layers or re-split depending on conversion status. In principle, this re-batching can add scheduling overhead. However, we find that:

- **KV Cache Savings Dominate.** Converting even a fraction of layers significantly reduces the cumulative KV cache, lowering overall memory pressure and boosting throughput in the rest of the pipeline.
- **Empirical Minimal Impact.** Our experiments (§5.4) show that the time spent splitting and merging batches is relatively small compared to the overall inference cost, especially for large prompt lengths.

Thus, although batched requests may follow diverging paths, the net system throughput remains high. Combined with our chained fine-tuning strategy, DSLA-*Serve* enables a flexible and efficient inference pipeline that handles heterogeneous requests while maintaining strong accuracy.

## 5. Experiments

**Experimental Setup.** In this section, we present the experimental analysis of the proposed method. We used 1.6 billion tokens sampled from the SlimPajama dataset (Shen et al., 2024) to fine-tune the model. We distilled our model from Llama2-7B (Touvron et al., 2023). For additional setup, please see Appendix B.

**Baselines.** We compare the adaptive distilled models generated via DSLA with representatives state-of-the-art from linear-complexity models (RetNet-6.7B (Sun et al., 2023b), GLA-7B-20B (Yang et al., 2023), Mamba-7B (Gu & Dao, 2023)), hybrid model (Zamba-7B (Glorioso et al., 2024)), and quadratic model (Llama2-7B (Touvron et al., 2023)), respectively. We report the performance of DSLA versions with 25%, and 50% layers converted to linear attention alternatives.

### 5.1. Long-Context Understanding

**Setup.** We evaluate our DSLA model on tasks requiring careful long-range interactions (Bai et al., 2024), including: (i) Multi-Document QA, where the model must answer questions using information spread across at least two documents; (ii) Code Understanding, which require reasoning across multiple code files to answer complex questions; (iii) Few-shot learning requires the model to comprehend a limited number of examples and effectively perform various tasks, including classification, summarization, and reading comprehension. Additionally, we report the perplexity on WikiText-2 and Lambada and compare the DSLA model, with 25% and 50% converted layers, against other models at 7B-scale. Training cost is reported in terms of the number of tokens processed.

**Results.** As shown in Table 1, our DSLA significantly outperforms other linear-cost models (GLA-7B, Mamba-7B)

and the hybrid model (Zamba-7B), achieving up to a **72.23%** performance improvement (*e.g.*, compared to Mamba-7B on TriviaQA). This substantial gain is likely attributed to the distillation process, which effectively transfers knowledge from the Llama-2-7B teacher model to the converted model. Notably, in multi-document QA tasks, our converted models *even surpass pure self-attention models*, achieving accuracy gains of 3.96 to 5.44 for the 25% converted model. Such conversion also offers practical benefits, reducing memory consumption by 1GB for a 4K context and 2GB for an 8K context per request.

Furthermore, we evaluate our DSLA on general language modeling tasks, reporting perplexity in Table 1, as well as on summarization tasks in Table 2. For summarization, we utilize the widely used CNN/DailyMail dataset (Hermann et al., 2015; See et al., 2017) and the XSum dataset (Narayan et al., 2018), reporting performance using ROUGE metrics (Lin, 2004). Our results indicate that DSLA effectively retains performance comparable to Llama2-7B and Zamba-7B while outperforming GLA-7B and Mamba-7B. This suggests that the DSLA retains the flexibility and adaptability needed for long-context understanding, all while providing substantial memory and computation savings.

### 5.2. Short-Context Benchmarks

Although the primary goal of DSLA is to enhance long-context understanding in efficient models, we also ensure that it maintains strong short-context performance without any degradation. Table 3 presents the performance of DSLA on seven common-sense reasoning tasks (lm_eval), where it demonstrates competitive performance against other baseline models, including pure self-attention models, hybrid and linear-complexity models. Additional results compared with an alternative linearization method (Zhang et al., 2024a) are provided in Table 7 in Section 5.5.

### 5.3. Inference Efficiency

In this section, we evaluate the inference latency and per-request memory usage during the prefill and decoding stages. Figure 4 shows the prefill latency and decoding latency normalized by generated length.

In the prefill stage, the full transformer shows lower latency for prompts under 2K tokens, while the converted model excels for prompts over 2K tokens. This improvement is attributed to the DSLA layer's design, which utilizes single linear layers for gating to minimize latency overhead while retaining the gating mechanism's benefits. Performance gains are particularly notable for context lengths up to 8K tokens.

During decoding, transformer latency increases with the length of generated tokens due to the self-attention mecha-

*Table 1.* Results of long context understanding. We report 25%, 50% converted layers of DSLA. DSLA is distilled from Llama2-7B.

| Methods | Cost | WikiText-2 ↓ | Lambada ↓ | Multi-doc QA | | Code Understanding | | Few-shot Learning | | |
| | | | | HotpotQA ↑ | 2WikiMQA ↑ | LCC ↑ | Repobench ↑ | TREC ↑ | Samsum ↑ | TriviaQA ↑ |
|---|---|---|---|---|---|---|---|---|---|---|
| Llama2-7B | 2T | **8.79** | 4.13 | 5.63 | 10.24 | **69.83** | **56.88** | 59.00 | **39.1** | 86.19 |
| GLA-7B | 20B | NaN | 4.98 | 3.61 | 6.89 | 41.26 | 44.24 | 28.50 | 16.94 | 57.68 |
| Mamba-7B | N/A | 10.55 | 4.05 | 1.23 | 0.80 | 17.56 | 10.54 | 11.0 | 4.55 | 15.23 |
| Zamba-7B | 1T | 10.25 | **3.74** | 7.90 | 7.97 | 40.70 | 43.20 | **64.0** | 37.74 | 82.19 |
| DSLA [25%] | 1.6B | 9.26 | 4.14 | **11.07** | **14.20** | 66.91 | 51.53 | 55.0 | 38.66 | **87.46** |
| DSLA [50%] | 1.6B | 9.89 | 6.19 | 10.61 | 13.64 | 61.68 | 49.84 | 46.0 | 37.28 | 81.99 |

*Table 2.* Comparison results on text summarization tasks.

| | CNN/DailyMail | | XSUM | | Avg. |
| | Rouge-1 ↑ | Rouge-L ↑ | Rouge-1 ↑ | Rouge-L ↑ | |
|---|---|---|---|---|---|
| Llama2-7B | 30.13% | 27.63% | 30.07% | 24.51% | 28.09% |
| GLA-7B | 18.55% | 17.62% | 24.65% | 20.92% | 20.44% |
| Mamba-7B | 22.13% | 21.04% | 27.92% | 23.49% | 23.65% |
| Zamba-7B | **31.34%** | **28.67%** | 28.41% | 23.09% | 27.88% |
| DSLA [25%] | 30.01% | 27.10% | **30.10%** | **24.70%** | 27.98% |
| DSLA [50%] | 29.29% | 26.42% | 29.88% | 23.73% | 27.33% |

*Table 3.* Comparison results on `lm_eval` tasks.

| | WG ↑ | HS ↑ | PIQA ↑ | ARC-E ↑ | ARC-C ↑ | MMLU ↑ | LogiQA ↑ | Avg.↑ |
|---|---|---|---|---|---|---|---|---|
| Llama2-7B | 69.3% | 57.1% | 79.6% | 76.4% | 43.3% | 45.3% | **30.6%** | **57.2%** |
| RetNet-6.7B | 66.1% | - | 77.8% | 73.3% | 39.9% | 26.1% | - | - |
| GLA-7B | 69.5% | 57.0% | 79.2% | 74.6% | **44.0%** | 28.4% | 28.8% | 54.5% |
| Mamba-7B | 71.9% | **58.6%** | **79.9%** | **77.7%** | 42.8% | 33.9% | 23.7% | 55.5% |
| Zamba-7B | **69.9%** | 57.0% | 78.8% | 74.2% | 42.2% | **48.9 %** | 27.0 % | 56.9% |
| DSLA [25%] | **69.9%** | 56.5% | 78.7% | 74.9% | 43.2% | 46.2 % | 24.9% | 56.3% |
| DSLA [50%] | 68.1% | 55.4% | 75.0% | 71.6% | 40.4% | 43.2 % | 25.4% | 54.2% |

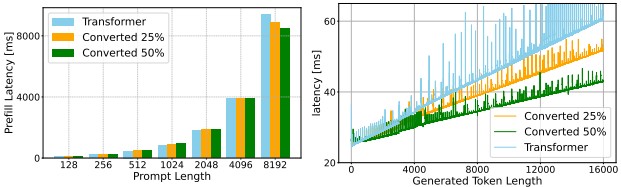

*Figure 4.* Prefill (left) and per-token decoding (right) latency.

nism's reliance on past token interactions, resulting in linear cost growth. Conversely, the converted model maintains a constant latency increase by updating only two states for historical and recent information, significantly reducing overall latency.

During decoding, we observed an interesting phenomenon: the latency fluctuation is significantly higher for the transformer model and decreased as we converted more layers into DSLA layers. Notably, the latency spikes consistently occurred at the same position (Appendix. E). Through profiling, we observed that memory allocation (`cudaMalloc`) occasionally took 300–1000 ms, while a single inference typically takes only 30–60 ms. Since this issue is related to memory allocation, the fluctuations reduce as we convert more self-attention layers to DSLA layers. This is because DSLA reduces memory usage by eliminating the necessity for a key-value (KV) cache.

DSLA avoids maintaining a KV cache, whose size grows linearly with the context length $s$ and the number of layers

$n$. The KV cache size per layer is given by:

$$ \text{KVCache}[B] = 2 \times n \times b \times s \times h \times d_h \times B_{byte} $$

Here, $n$ is the number of layers, $b$ is the batch size, $s$ is the context length, $h$ is the number of heads, $d_h$ is the dimension per head, and $B_{byte}$ is for the data type size (e.g., 2 for BF16, 8 for FP64). Thus reducing the number of transformer layers significantly reduces memory usage.

### 5.4. End-to-end System Performance

In this section, we evaluate the end-to-end system performance by simulating a real-world workload using augmented Azure inference traces (Appendix. C). To simulate this workload, we replayed user requests from these traces on DSLA-*Serve* and measured the normalized end-to-end latency of the requests.

By detecting the load, based on the prompt lengths of requests seen, the DSLA-*Serve* automatically adjusts the conversion rate and progressively converts self-attention layers to DSLA layers, ranging from 25% to 50%. Table 4 presents the prompt length distribution from the trace and the corresponding conversion rates we applied.

The result shows a significant improvement in the system's efficiency. The average end-to-end latency, normalized by token length, improved by **2.29**×, demonstrating the effectiveness of DSLA-*Serve* in adapting to dynamic system load while maintaining performance.

*Table 4.* Prompt length distribution and maximum conversion rate set during the experiment. DSLA-*Serve* improves end-to-end per-token generation latency by 2.29×.

| Prompt Length | Distribution | Max Conv. Rate |
|---|---|---|
| seq_len<2k | 64.68% | 12.5% |
| 2k≤seq_len<4k | 16.16% | 25% |
| 4k≤seq_len<8k | 16.03% | 37.5% |
| 8k≤seq_len | 3.1% | 50% |
| Latency (before) | 93.64ms | |
| Latency (after) | 40.83ms | |

## 5.5. Further Investigation and Ablation Studies

**Importance of specialized hidden states.** As discussed in §3, the primary limitation of linear attention lies in its tendency to focus on the most recent tokens while neglecting earlier ones. In Figure 5, we compare the attention patterns of standard self-attention (left) and DSLA (right). Unlike the single-state GLA (Fig.1), the *recency state* and *history state* in DSLA are specialized to attend to different regions of the input. This specialization enables the history state to effectively capture information from earlier tokens.

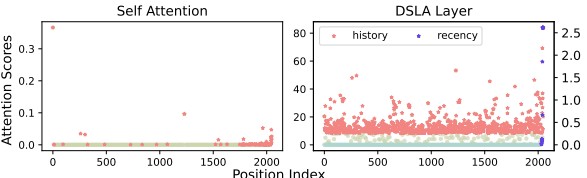

*Figure 5.* Attention score analysis.

**History vs recency per layer.** We visualize the $\gamma$ value used as a weighting factor that determines how the two hidden states contribute to the output ($\mathbf{o}_t = \mathbf{q}_t(\gamma \cdot \mathbf{S}_t^1 + (1-\gamma) \cdot \mathbf{S}_t^2)$) in Figure 6.

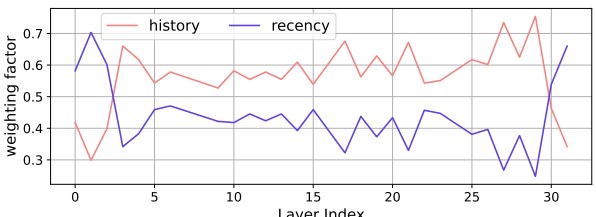

*Figure 6.* Hidden states weighting factor visualization.

In the figure, the state with the higher value dominates each layer. Specifically, the *recency state* is predominant in the initial layers (0–3) and the final layers (30–31), whereas the *history state* takes precedence in the middle layers (4–29).

This observation aligns with previous research emphasizing the distinct roles played by different layers in a model (Jawahar et al., 2019; Liu et al., 2024a). Fine-grained control across layers can significantly enhance language modeling performance. By learning the parameter $\gamma$ on a layer-by-layer basis, the model dynamically adjusts the relative importance of the two hidden states at each layer. This adaptability enables the model to focus on specific attention patterns or temporal dependencies as required by each layer, ultimately improving overall performance.

**Is dual-state enough?** We conduct an ablation study to assess the impact of the number of hidden states in GLA. These experiments are performed using Llama-3.2-1B-Instruct on the CoLA task (Wang, 2018). Specifically, we first perform supervised fine-tuning on CoLA, then replace the self-attention layers with GLA layers while vary-

ing the number of hidden states. The resulting models are re-trained on the CoLA task to recover performance. For configurations with more than two states, we compute the $\mathcal{L}_{cont}$ between consecutive states and average the values.

*Table 5.* Ablation study of the number of hidden states. Results are reported in accuracy on the validation set.

| # States | 1 | 2 | 4 | 8 | Baseline |
|---|---|---|---|---|---|
| Llama-3.2-1B-Instruct | 78.97 | 81.19 | 81.44 | 80.96 | 82.83 |

The results, presented in Table 5, show that our DSLA significantly improves the ability to replicate the original self-attention mechanism, achieving a 2.22 accuracy increase over the single-state GLA. Moreover, increasing the number of hidden states beyond two yields diminishing returns, as the dual-state setup is sufficient to effectively capture both historical and recent information.

**Comparison with hybrid architecture.** We measured the latency of Zamba-7B to compare its efficiency. For a fair evaluation, we used the author-provided checkpoint and code, adhering to the instructions for environment setup. On average, Zamba-7B is 1.8× *slower* in prefill across all tested context lengths, 3.0× *slower* in end-to-end (including generation) latency than the transformer-based architecture Llama2-7B on an A100 machine. This performance gap is primarily attributed to Zamba-7B's extensive number of activated parameters and self-attention layers. Specifically, Zamba-7B employs 13 shared self-attention layers that are repeatedly loaded and utilized, exacerbating the memory bottleneck unless executed on state-of-the-art hardware such as the H100 (used in the original paper (Glorioso et al., 2024)), which offers 1.5× higher memory bandwidth compared to A100 that we used (H100; A100).

**Effectiveness of sensitivity metric.** Here, we discuss the effectiveness of attention entropy (§4.3) as a sensitivity metric. For the ablation study on other potential metrics (outlier percentage, and downstream task accuracy), please refer to Appendix F. Figure 7 visualizes the correlation between the metric and its performance impact, where the X-axis represents the layer index at which a self-attention layer is replaced with a DSLA layer, and the primary Y-axis shows the resulting performance impact, measured as perplexity on the WikiText-2 dataset. The metric is plotted on the secondary Y-axis. A strong correlation between a metric and performance impact suggests its usefulness in guiding model modifications.

**Comparison with a different linearization method** In Table 7, we compare our method with LoLCATs (Zhang et al., 2024a) on `lm_eval` tasks. For a fair comparison, we applied our approach to the same teacher model (Llama3-8B) used by LoLCATs. While LoLCATs leverages low-rank adaptation to recover performance lost during linearization,

*Table 6.* Results of long context understanding. We report 25%, 50% converted layers of DSLA. DSLA is distilled from Llama2-7B-chat, an instruction tuned model.

| Methods | Cost | Multi-doc QA | | Code Understanding | | Few-shot Learning | | |
|---|---|---|---|---|---|---|---|---|
| | | HotpotQA ↑ | 2WikiMQA ↑ | LCC ↑ | Repobench ↑ | TREC ↑ | Samsum ↑ | TriviaQA ↑ |
| Llama2-7B-chat | 2T | 29.79 | 27.15 | **59.71** | **48.96** | 58.33 | 39.03 | 86.11 |
| GLA-7B | 20B | 3.61 | 6.89 | 41.26 | 44.24 | 28.50 | 16.94 | 57.68 |
| Mamba-7B | N/A | 1.23 | 0.80 | 17.56 | 10.54 | 11.0 | 4.55 | 15.23 |
| Zamba-7B-Instruct | 1T | 24.15 | 29.96 | 47.06 | **48.96** | **70** | **40.35** | **88.89** |
| DSLA [25%] | 800M | **35.37** | **32.33** | 58.11 | 45.17 | 55.0 | 40.07 | 85.56 |
| DSLA [50%] | 800M | 33.28 | 30.71 | 53.55 | 41.75 | 44.0 | 37.64 | 73.04 |

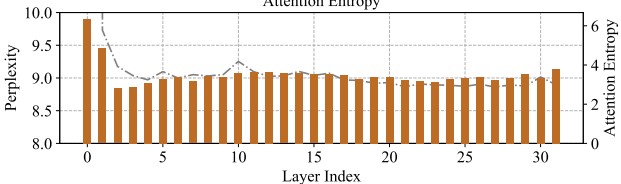

*Figure 7.* Attention entropy as a sensitivity metric.

DSLA employs a dual-state mechanism to separately retain historical and recent information. Additionally, DSLA-*Serve* enables dynamic adaptation to accuracy-latency trade-offs. On average, DSLA achieves superior performance. We report the standard deviation of our measurements for reference.

*Table 7.* Comparison of different linearization methods. DSLA is distilled from Llama-3 8B.

| | WG ↑ | HS ↑ | PIQA ↑ | ARC-E ↑ | ARC-C ↑ | MMLU ↑ | Avg. ↑ |
|---|---|---|---|---|---|---|---|
| Llama3-8B | 74.1% | **79.7%** | 79.9% | 80.1% | 53.3% | 66.6% | 72.2% |
| LoLCATs | **80.9%** | **79.7%** | **80.9%** | **81.7%** | 54.9% | 52.8% | 70.7% |
| DSLA | 73.2%±1.3 | 77.5%±0.5 | 79.8%±1.0 | 80.7%±0.8 | **55.9%**±1.5 | **68.9%**±3.2 | **72.7%** |

**Application to different models** We evaluated the proposed method on a smaller-scale model (1.5B) to demonstrate its scalability. Specifically, we distilled the Microsoft Phi-1.5 (Li et al., 2023) model and compared it with the baseline Transformer architecture, the 1.5B SSM model Phi-Mamba (Bick et al., 2024), and our hybrid model. Consistent with the results on larger models (Table 1–Table 3), our hybrid model outperforms all baselines. These results confirm that our method generalizes well across different model scales.

*Table 8.* Comparison of 1.5B-scale models on `lm_eval` tasks. DSLA is distilled from Phi-1.5.

| | Architecture | Cost | WG↑ | ARC-E↑ | ARC-C↑ | PIQA↑ | HS↑ |
|---|---|---|---|---|---|---|---|
| Phi-1.5 | Transformer | 150B | 73.4 | 75.0 | 48.0 | 76.6 | 62.6 |
| Phi-Mamba | SSM | 3.0B | 71.7 | 74.0 | 44.1 | 75.5 | 60.2 |
| DSLA | DSLA | 800M | **72.9** | **74.5** | **44.5** | **75.9** | **60.5** |

We provide additional results in Table 6 to demonstrate the generalizability of our method across different architectures.

Specifically, Table 6 reports results using the Llama-2-7B-chat model. The main observations discussed in §5.1 remain consistent, confirming that our method performs robustly across varying backbone models.

**Limitations** To enable fast inference without adding model loading latency to the critical path, DSLA-*Serve* loads both architectures (i.e., Transformer layers and DSLA layers) into memory. This approach may require additional memory to store the model weights. To mitigate this, we can consider offloading layers and *prefetching* layers, as the layer replacement order is known in advance. Offloading and prefetching of weights at inference time have been explored in prior work (Aminabadi et al., 2022; Cai et al., 2024a; Eliseev & Mazur, 2023), and are regarded as effective techniques for reducing memory usage—especially if the server overlap prefetching with computation or communication, thereby hiding loading latency.

In terms of accuracy, we empirically observed that performance begins to degrade when more than 75% of the model is converted, consistent with observations from (Wang et al., 2024a). However, DSLA-*Serve* dynamically adjusts the conversion rate to meet the service level agreement (SLA). In general, the performance of a distilled model can be improved by employing a stronger teacher model, as demonstrated in Table 6 and Table 7.

## 6. Conclusion

In this work, we proposed DSLA-*Serve*, an adaptive inference system that distills quadratic complexity self-attention layers to linear-cost DSLAlayers to reduce inference cost *on the fly*. For the linear-cost model, we use DSLA a dual-state linear attention highlighting two hidden states that separately hold history and recent context. With the specialized states, DSLA overcomes the limitations of previous linear attention models, which primarily focus on localized context. With DSLA-*Serve* we achieve 2.3× end-to-end improvement over the transformer model and 3.0× reduction over the hybrid model while maintaining competitive performance on downstream tasks.

## Acknowledgements

We thank the anonymous reviewers for their constructive feedback. Ro and Akella are supported by NSF grants CNS-2105890 and CNS-2232135, as well as by Cisco Research and Meta. Ro is also supported by IBM PhD Fellowship. This work was supported in parts from Intel (single PI SRS grant).

## Impact Statement

This work aims to improve the efficiency of inference of LLMs. It has no particular negative social or ethical impact beyond typical efficient LLM serving systems.

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

# A. System Overview

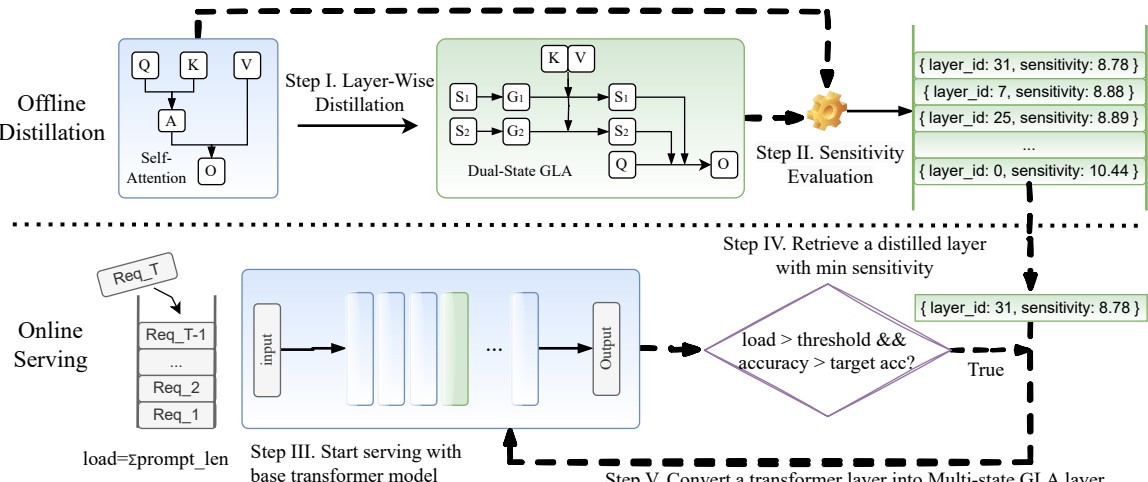

*Figure 8.* **Overview of DSLA-*Serve*.**

We provide a visual overview of DSLA-*Serve* in Figure 8.

We first perform a layer-wise fine-tuning (distillation) offline (*Top*), generating DSLA substitutes for each Transformer layer and ranking layers by their sensitivity to conversion.

During inference (*Bottom*), DSLA-*Serve* automatically replaces layers in ascending order of sensitivity as resource pressure increases, allowing the system to adaptively reduce memory and compute overhead with minimal accuracy loss.

# B. Detailed Experimental Setup

In this appendix, we provide additional details on the fine-tuning process of the DSLA model. We initialize the query, key, value, and output projection layers by reusing their pre-trained parameters. The history gating layer is initialized to be close to the identity matrix, while the recency gate is randomly initialized using $\mathcal{N}(0, \sigma^2)$. The biases of the gating layers are set to zero.

For fine-tuning, we use the Slimpajama dataset (Shen et al., 2024), which comprises Arxiv (Clement et al., 2019), Books (Gao et al., 2020), Common Crawl, C4 (Raffel et al., 2019), Github, Wikipedia (Foundation), and StackExchange (Excahnge, 2024). We randomly sample from the first chunk of the training dataset, truncating inputs to a maximum of 4K tokens. Tokenization is performed using LLaMA 2-7B's tokenizer (Touvron et al., 2023).

Fine-tuning is conducted using the AdamW optimizer (Loshchilov, 2017), with a weight decay of 0.01. The learning rate is linearly warmed up to 1e-4 over the first 5% of training steps, followed by a cosine decay schedule down to 5e-5. We leverage an open-source Triton kernel for fine-tuning (Yang et al., 2023).

For fine-tuning the 7B model and measuring latency (Fig.4), we use A100 GPUs with 80GB of memory. End-to-end performance evaluation (Table4), downstream task evaluations (Table 1-3), and ablation studies (Table 5) are conducted on A6000 GPUs with 49GB of memory.

In terms of fine-tuning cost, fine-tuning a single layer of a 7B model on 1B tokens takes approximately 5 hours on 4×A100 GPUs (80GB). Compared to full pretraining (858 days (Touvron et al., 2023)), our fine-tuning costs just 0.07%—a one-time expense.

# C. Serving Trace Information

To replay real-world serving scenarios, we augmented Azure LLM inference trace 2023 (Patel et al., 2024). The original trace contains a timestamp, the number of context tokens (prompt tokens), and the number of generated tokens. Since the original dataset does not have a text prompt and session, we generated a text prompt and added a session ID to enhance the

original dataset and use it to replay the real-world chat serving scenario. Figure 9 shows the distribution of chats by number of turns per session and the number of requests per second.

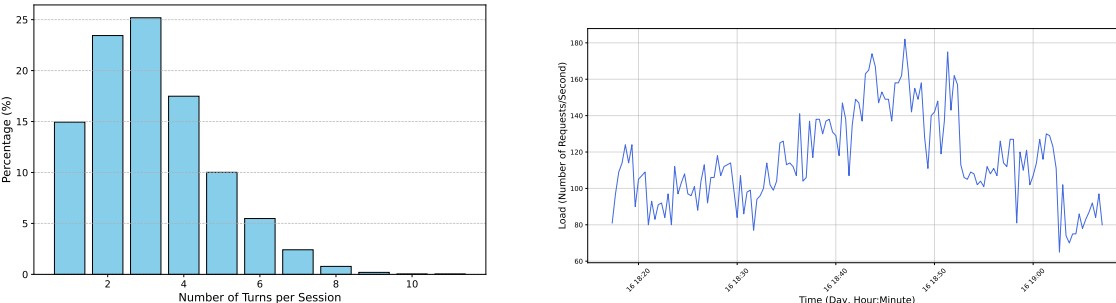

*Figure 9.* (a) Number of turns per session follows Poisson distribution ($\lambda = 3$). (b) Varying load defined as the number of user requests in real world serving scenario. Across time, the load to the server fluctuates, indicating the need to adaptive de-stressing solution for a server.

When we are replaying the conversation, we concatenate history of the session to simulate the multi-turn chat. By doing this, resulting trace's prompt length shows following distribution.

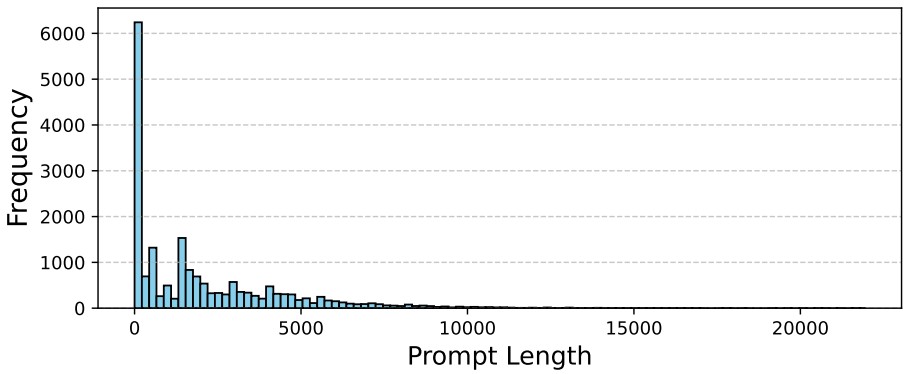

*Figure 10.* Prompt length distribution

## D. Serving of Mixed Batch

Assume there are four requests waiting in the queue from R1 to R4 (Figure 11). These are from different users. Due to differences in prompt length or other factors, the model used at the previous generation step for each request may differ. For instance, R2 with a short prompt used a transformer, while R4 with a very long prompt used a 3-layer-converted model.

If we now want to batch two generation requests, R1 and R2, together for the next token generation, a path divergence could occur at Layer 2. R1 needs Dual-state GLA (Layer 2-1), and R2 needs Self Attention (Layer 2-2). The diverged path from different needs resembles a mixture-of-experts architecture. As explained in the main manuscript, this can cause a delay because it cannot be batched for the operations in Layer 2. However, requests R1 and R2 can still be batched together at Layer 1, Layer 3, and Layer 4, thereby maximizing throughput. Additionally, the memory savings achieved at Layer2-1 by conversion are substantial enough to compensate for the delay.

User request used in the Figure 11 is from Chatbot Arena Dataset (Zheng et al., 2023).

## E. Latency Fluctuation

To analyze the fluctuation in decoding latency, we ran 2048 token generation tasks multiple times with the same prompt. Interestingly, we observed that the latency fluctuation occurs in the same location across multiple runs. With profiling with

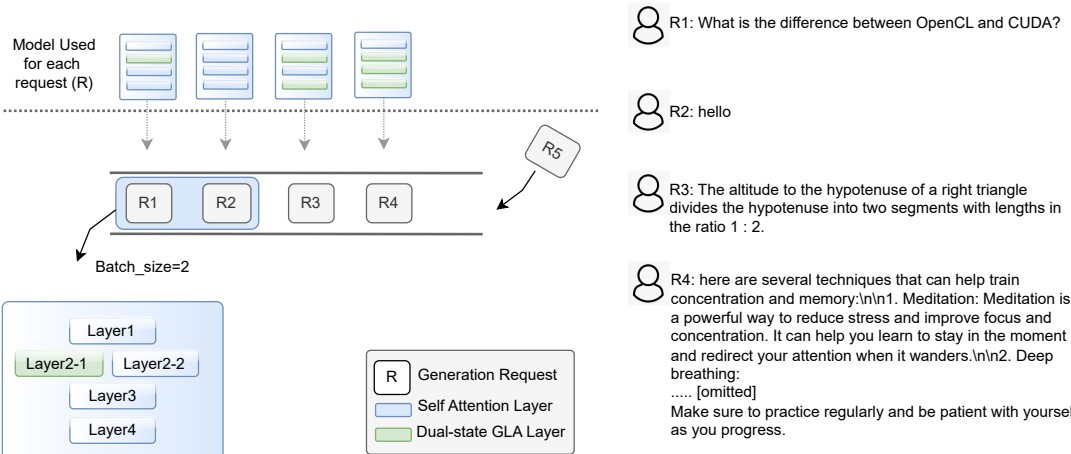

*Figure 11.* Serving of Mixed Batch

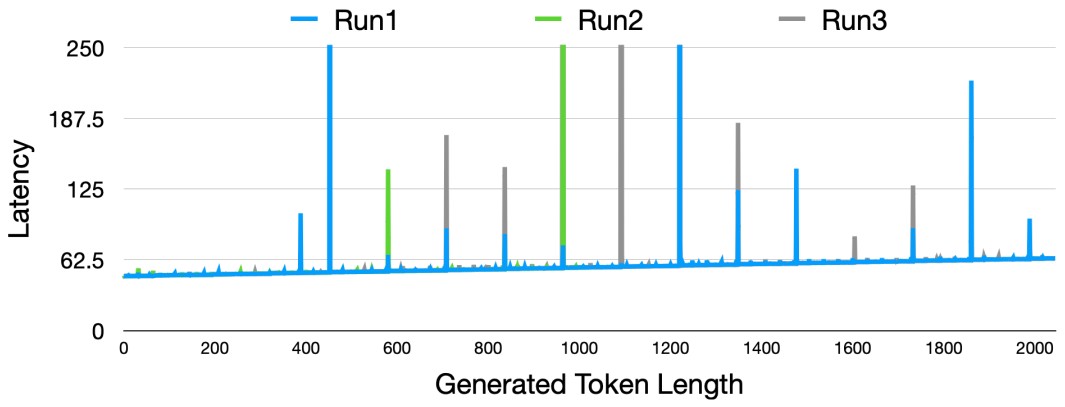

*Figure 12.* Latency fluctuation

pytorch profiler, we observed that this is due to the `cudaMalloc` operation, which takes 300ms-1000ms intermittently. This unexpected delay can be mitigated by converting more transformer layers into linear layers, which reduces the memory usage by removing the KV cache.

For this experiment, we used an A6000 GPU and a llama2-7b model, so the order of latency could differ from the numbers reported in the main manuscript.

## F. Further Discussion on Sensitivity Metric: An Ablation Study

In this section, we compare three sensitivity metrics (§4.3): (a) attention entropy, (b) outlier percentage, and (c) downstream task accuracy. The goal is to determine whether these metrics correlate with performance impact, making them effective sensitivity measures.

For the interpretation of figures, each sensitivity metric is plotted on the secondary Y-axis. A strong correlation between a metric and performance impact suggests its usefulness in guiding model modifications.

Other than attention entropy that we used, downstream task accuracy in Figure 13(c) worth further discussion. While the downstream task accuracy provides a more direct measure of sensitivity, it can be only known after the distillation is done so it could be less practical.

Additionally, its utility varies across tasks, making it less generalizable when a model is used for multiple applications.

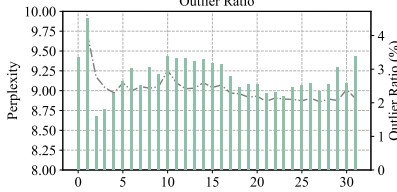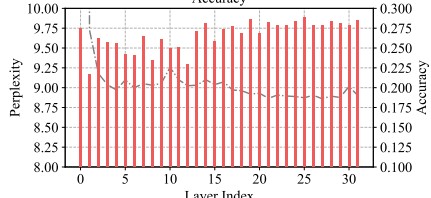

*Figure 13.* Different possible sensitivity metrics: (a) attention entropy, (b) outlier percentage, and (c) downstream task accuracy. We also observed that attention entropy tends to perform reasonably well across a variety of downstream tasks.

Figure 13(c) illustrates this limitation: the bar plot represents Wikitext perplexity, while the line plot shows downstream task performance (Rouge-1 score for summarization). These metrics do not always align, suggesting that task-specific accuracy is not always a suitable proxy for sensitivity. However, if a model is fine-tuned for a single task, using task accuracy as a sensitivity metric can be advantageous and improve performance (Ro et al., 2024).

## G. Additional Related Work

### G.1. Efficient Inference Techniques

Large language models (LLMs) are known for their high deployment costs, primarily due to the substantial parameter counts and the increasing overhead of the KV cache during generative decoding. To mitigate the parameter-related overhead, numerous studies have been conducted. Broadly, some research has focused on developing smaller models with carefully optimized training recipes (Liu et al., 2024b), while others have addressed parameter reduction through sparsity (Frantar & Alistarh, 2023; Ma et al., 2023; Sun et al., 2023a) and quantization (Frantar et al., 2022; Xiao et al., 2023a; Lin et al., 2024). To reduce the KV cache cost, which is often memory-bound during the iterative generation process, a series of works have focused on KV cache compression techniques (Zhang et al., 2023; Li et al., 2024; Tang et al., 2024) to alleviate memory I/O limitations. Additionally, other research has explored optimizing the decoding process itself, such as speculative decoding (Sun et al., 2024) and multi-head decoding (Cai et al., 2024b). In contrast to these efforts, this work investigates elastic inference through layer-wise conversion of sub-quadratic attention to support long-context processing.

### G.2. Knowledge Distillation

Knowledge distillation (Hinton, 2015) is an effective technique for transferring knowledge from large models to smaller ones, improving inference efficiency. To address the quadratic complexity of transformers, several initial efforts have focused on distilling transformers into linear attention models (Mercat et al., 2024; Kasai et al., 2021). Recently, (Bick et al., 2024) proposed a stage-wise distillation pipeline that effectively distills transformer-based Phi-3 models into a hybrid of transformer and Mamba models, with only a slight performance degradation. In addition, (Wang et al., 2024a) further improved efficiency by integrating speculative decoding. (Zhang et al., 2024b) explored the expressiveness of linear attention and introduced a simple polynomial approximations to enhance its capabilities. Previous approaches typically used knowledge distillation as a plug-and-play technique, successfully distilling quadratic transformers into sub-quadratic linear attention models. However, in this work, we systematically analyze the layer-wise and head-wise properties of the misalignments in attention patterns between self-attention and linear attention. And then we propose a simple and effective scaling strategy to mitigate the suboptimal distillation process caused by these misalignments.

