# OpenReview forum: "On-the-Fly Adaptive Distillation of Transformer to Dual-State  Linear Attention for Long-Context LLM Serving"
_ICML.cc/2025/Conference — ICML 2025 poster_

### Official Review · Reviewer_Gcze · 2025-03-07

**Overall Recommendation:** 3

**Summary:**

This work intends to reduce LLM inference complexity without unacceptable serving quality degradation. The contributions are two folds: (1) dual-state linear attention (DSLA): a variant of gated linear attention with two hidden states for history and recency (2) DSLA-Serve: an online adaptive distillation framework that uses an offline chained fine-tuning recipe and an inference time Transformer layer to DSLA layer replacement strategy guided by a sensitivity-based layer ordering. Evaluations on commonsense reasoning, long-context QA, and text summarization demonstrate that DSLA-Serve yields 2.3× faster inference than Llama2-7B and 3.0× faster than the hybrid Zamba-7B, while retaining comparable performance across downstream tasks

**Claims And Evidence:**

1. Claim: DSLA can preserve historical context, which overcomes the short-range bias observed in single-state GLA.

Evidence: Figure 5 shows that the recency state and history state in DSLA are both effective.

2. Claim: Attention entropy is an effective sensitivity metric.

Evidence: Figure 7 shows a strong correlation between the attention entropy of each layer and its performance impact. Appendix F also compares other metrics.

3. Claim: DSLA-Serve achieves significant inference speed-ups while maintaining competitive task performance compared to both full Transformer and hybrid models.

Evidence: Section 5 (supported by Tables 1–3 and Figure 4) reports that the adaptive model runs 2.3× faster than Llama2-7B and 3.0× faster than

**Essential References Not Discussed:**

I didn’t identify essential references not discussed in the paper.

**Experimental Designs Or Analyses:**

I have checked the soundness of the experimental designs and analyses.
1. Section 5.1 evaluates the DSLA models of two conversion rates on perplexity and the performance on long-context understanding.
2. Section 5.2 evaluates the DSLA models on short-context benchmarks
3. Section 5.3 evaluates the inference latency and memory usage during prefill and decoding stages. However, it is unclear what the prompt length is for the decoding latency experiment.
4. Section 5.4 evaluates the end-to-end system performance on open-source LLM serving traces.
5. Section 5.5 does ablation study on the ratio between history and recency, number of states, efficiency compared with Zamba-7B, and the effectiveness of sensitivity metric.

**Methods And Evaluation Criteria:**

The proposed methods and evaluation criteria are well-tailored to the problem. The authors use a broad range of benchmark datasets (e.g., Multi-Document QA, Code Understanding, WikiText-2, Lambada, and summarization tasks like CNN/DailyMail and XSum) that stress both long-context understanding and short-context performance. Hidden states weighting factor clearly shows the tradeoff between history and recency. LLM serving metrics such as memory usage and latency show the system advantage.

**Other Comments Or Suggestions:**

I don’t have other comments or suggestions.

**Other Strengths And Weaknesses:**

The experiments are comprehensive. However, the effectiveness of sensitivity is unclear after all the post-training stages including RL.

**Questions For Authors:**

1. What LLM serving system are you using? Could you explain why shorter generation length can take longer time in the autoregressive process? GPU memory allocation can explain the burst latency but cannot explain my concern.
2. If I understand correctly, the KV cache of the replaced layer is removed without being incorporated into the hidden states of the new DSLA layer. Have you tried to use the KV cache to further improve the model quality?

**Relation To Broader Scientific Literature:**

DSLA provides new insights to modeling memory and forget mechanisms. DSLA-Serve provides a new recipe for adaptive inference, which is key to understanding the extremely efficient human reasoning process.

**Theoretical Claims:**

There are no theoretical claims to check.

---

> ### Author Rebuttal · Authors · 2025-04-01
>
> Thank you for the detailed and constructive comments! Please see below.
>
> **A1. Serving system details**
> We implemented our inference system on top of the DeepSpeed inference serving framework (MII). In the autoregressive process, for relatively short prefill lengths, our method may be slower than Transformers due to the additional linear projection layers. In addition to the standard K, Q, V, and O projections, DSLA introduces two extra layers to compute G1 and G2 (see Eq. 4 in our paper). This overhead impacts latency when memory is not the bottleneck. However, as we generate more tokens, GPU memory becomes the main bottleneck, and our method begins to outperform Transformers in terms of latency.
>
> **A2. KV cache**
> Thank you for the question. To clarify, the KV cache is not entirely “removed” but rather “implicitly integrated” into the hidden states of the DSLA layers. In the original attention layers, the full KV cache is retained, where each KV pair can be considered a hidden state, and the output depends on all such states. In contrast, the DSLA layer maintains only two hidden states, into which the information from all KV pairs is incrementally aggregated (please check the equation 4). This design significantly reduces the memory footprint, thereby improving efficiency.
>
> **A3. Effectiveness on post-trained models**
> Thanks for the interesting question! To evaluate our method's effectiveness on reasoning tasks, we tested it on the Qwen/Qwen2.5-Math-1.5B-Instruct model, which is fine-tuned with SFT followed by RL using GRPO. During distillation, we used open-source reasoning datasets [1,2], distilled 10% of the layers, and evaluated them on GSM8K and AIME24. Our model matched the teacher’s accuracy on GSM8K, showing that our method works on post-trained models.
>
> Table A.
> | Model                           | GSM8K |
> |---------------------------------|-------|
> | Qwen/Qwen2.5-Math-1.5B-Instruct | 85.0  |
> | DSLA-Qwen2.5-Math-1.5B-Instruct | 84.9  |
>
> Reference.
>
> [1] https://github.com/huggingface/open-r1
>
> [2] https://huggingface.co/datasets/HuggingFaceH4/numina-deepseek-r1-qwen-7b
>
> [3] Muennighoff, Niklas, et al. "s1: Simple test-time scaling." arXiv preprint arXiv:2501.19393 (2025).

---

### Official Review · Reviewer_a2iq · 2025-03-14

**Overall Recommendation:** 4

**Summary:**

This work presents a robust approach to deploying gated linear attention (GLA) in practical production environments, addressing key limitations through two primary innovations.

First, the authors address GLA’s strong recency bias, which impairs long-context performance. To mitigate this, they propose a dual-state mechanism. One hidden state models local context with a randomly initialized forget gate, while the other models long-term dependencies with a forget gate initialized closer to one (preserving earlier information). By introducing data-dependent dynamic interpolation and a contrastive penalty loss, the model effectively balances reliance between these two states.

Second, the paper proposes DSLA-Serve, an adaptive distillation framework that selectively converts Transformer layers into DSLA layers during inference. To guide this conversion, the authors introduce an entropy-based sensitivity metric that identifies less critical layers. The framework dynamically substitutes these layers based on system conditions, ensuring performance stability. The chained fine-tuning strategy is introduced to maintain consistency after layer conversion.

The experimental results demonstrate that DSLA effectively mitigates recency bias and achieves competitive performance on benchmarks, offering improved efficiency compared to other hybrid models. Additional discussions cover batched inference strategies for further deployment optimization.

**Claims And Evidence:**

Figures 3 and 5 provide evidence that the dual-state mechanism effectively enhances long-context capabilities.

Table 4 demonstrates that converting multiple layers to linear attention can maintain performance.

Regarding the efficiency claim, the evaluation is limited to comparisons with Llama2-7B and Zamba-7B, omitting a comparison with other hybrid architectures such as NVIDIA's 7B hybrid Mamba2.

**Essential References Not Discussed:**

No

**Experimental Designs Or Analyses:**

Yes, the experimental design and analyses appear generally sound.

Ablation studies are relatively thorough, but the impact of chaining order during fine-tuning is not fully explored.

**Methods And Evaluation Criteria:**

Yes, the proposed methods and evaluation criteria are appropriate for the problem at hand. The paper evaluates performance on both short- and long-context benchmarks, complemented by efficiency comparisons, aligning well with the intended application.

**Other Comments Or Suggestions:**

No

**Other Strengths And Weaknesses:**

Overall, I believe this is a solid industrial-style paper that aims to improve efficient long-context modeling. If ICML is receptive to such contributions, I would recommend acceptance.

However, the paper lacks comparisons with key baselines, particularly other distilled hybrid models such as LOLCATs, Transformers to SSMs, and Mamba-in-LLaMA. Including these comparisons would provide a more comprehensive evaluation of the proposed method’s effectiveness.

**Questions For Authors:**

Have you explored more recent linear attention variants, such as Gated DeltaNet [ICLR '25]? It would be interesting to investigate whether the proposed dual-state technique is applicable to other types of linear attention. Demonstrating its generalizability could significantly strengthen the paper’s contribution.

**Relation To Broader Scientific Literature:**

This paper leans more toward an industrial focus; however, the proposed dual-state mechanism presents a meaningful methodological contribution to long-context modeling in linear attention. The dual-state design effectively addresses recency bias, a known limitation in linear attention models, by combining specialized hidden states for local and long-term dependencies. This innovation aligns with broader research on improving linear attention mechanisms and contributes to the ongoing effort to enhance efficiency in hybrid softmax/linear attention architectures. By enabling more effective deployment of such hybrids in long-sequence settings, this approach opens new opportunities for scalable long-context modeling.

**Theoretical Claims:**

This is not a theoretical paper.

---

> ### Author Rebuttal · Authors · 2025-04-01
>
> Thank you for the thoughtful and constructive comments! Please see below.
>
> **A1. Comparison to Other Baselines**
> Thanks for the great suggestions. We compare our 1.5B-scale model with the Phi-Mamba-1.5B [1] and 7B-scale model to distilled Mamba [2]. As demonstrated in table A and B, our method consistently outperforms these baselines as well.
>
> For the 1.5B-scale model, we used the same teacher model– Microsoft's phi-1_5 model and used 800M tokens. We used the same environment setting and dataset reported in the main draft.
>
> Table A.
>
> | Method                   | Token | Winogrande | ARC-e     | ARC-c     | PIQA      | Hellaswag |
> |---|---|---|---|---|---|---|
> | Transformers to SSMs [1] | 3.0B  | 71.7       | 74.0      | 44.1      | 75.5      | 60.2      |
> | DSLA (Ours)              | 800M  | **72.93**  | **74.49** | **44.45** | **75.90** | **60.48** |
>
>
> For the 7B-scale model, note that we used Llama2-7B while [2] used Llama3 to start with.
>
> Table B.
>
> | Method | PiQA | ARC-c | Hellaswag | MMLU | Winogrande |
> |---|---|---|---|---|---|
> | distilled Mamba [2] | **78.7** | **52.4** | **77.7** | 42.4 | 64.8 |
> | DSLA (Ours) | **78.7** | 42.8 | 74.1 | **48.5** | **67.8** |
>
> Also, thanks for bringing up Nvidia’s 8B Mamba Hybrid model. It's an interesting baseline for hybrid architectures. We chose Zamba-7B as one of the representative hybrid models because it is an open-source model of similar scale (7B) that can run on our machines. While both Zamba-7B and Nvidia’s Hybrid Mamba share similarities in terms of model architecture, both have limitations due to its fixed architecture, whereas DSLA can dynamically adapt to variability of load during inference.
>
>
> **A2. Generalization: Application to a different linear architecture**
> Thanks for the great suggestion! Unfortunately we were not able to get the parameters for GatedDeltaNet as it was not public. Instead, we applied the proposed method to an alternative sub-quadratic model, Mamba, to demonstrate its generalizability. Mamba shares similarities with GLA that we picked in the paper due to the presence of "selective gate". We extended Mamba’s selective gate to dual gate and fine-tuned the gate parameters with 400M tokens using the same setting used in our paper. Due to the time constraints, we were only able to convert 25% of layers. Table C shows a zero-shot performance of single state Mamba and dual state Mamba.
>
> Table C.
>
> | Model                  | Winogrande | ARC-e    | ARC-c    | PIQA     | Hellaswag |
> |---|---|---|---|---|---|
> | Mamba-1.3B             | 54.1       | 59.0     | 28.2     | 72.2     | 40.1      |
> | Dual-state Mamba [25%] | **54.4**   | **60.1** | **28.7** | **72.2** | **47.5**  |
>
> This highlights the generalizability of the DSLA method across various architectures.
>
> Reference.
>
> [1] Bick, Aviv, et al. "Transformers to ssms: Distilling quadratic knowledge to subquadratic models." Advances in Neural Information Processing Systems 37 (2024): 31788-31812.
>
> [2] Wang, Junxiong, et al. "The mamba in the llama: Distilling and accelerating hybrid models." Advances in Neural Information Processing Systems 37 (2024): 62432-62457.

---

### Official Review · Reviewer_RVY3 · 2025-03-14

**Overall Recommendation:** 3

**Summary:**

This paper introduces an online distillation framework that dynamically converts Transformer layers to dual state linear attention (DSLA) during inference to improve efficiency for long-context LLM serving. DSLA uses two specialized hidden states to better preserve both historical context and recent information, addressing the short-range bias typical in linear attention architectures. Through a sensitivity-based layer conversion ordering and chained fine-tuning strategy, DSLA-Serve achieves 2.3-3.0× faster inference compared to baselines while maintaining comparable accuracy across various downstream tasks.

**Claims And Evidence:**

Well supported claims:
1. *Dual-state architecture*: The claim that it maintains both historical and recent context is supported through ablation analysis compared to single-state architecture and attention/weighting factor visualizations.
2. *Framework adaptability*: The framework is shown to be adaptive on a real-world workload, demonstrating that the conversation rate is adapted to the prompt length which leads to about 2.27x reduction in latency.
3. *Inference speed*: The authors provide evidence for the greater than 2x speed claims using two different base large language models, Llama2 and Zamba 7B models. In addition, they show latency improvements in a real-world workload scenario.

Partially supported claims:

4. *Comparable quality to baselines*: The results do support that results are comparable on average but the individual results are sometimes worse and are dependent on the conversion rate. This could be problematic as it may non-trivial to find a conversion rate that works well across all datasets.

**Essential References Not Discussed:**

See my reply above.

**Experimental Designs Or Analyses:**

The experimental design has been well-designed for the most part: baseline comparisons, evaluation task diversity, ablation studies, real-world workload results. There a few parts that can be improved in the exposition and analysis:

1. Adding some discussion on the training overhead that is introduced when training the dual-state attention.
2. It would be useful to experiment with different deployment scenarios to identify in which ones the proposed approach is most beneficial.
3. A few technical details that are missing from the experiment section such as what inference optimization techniques were used and what are the details of the inference workflow used by each method.
4. The experimental analysis would benefit by showing some statistical error information to better understand the performance differences and how consistent they are.

**Methods And Evaluation Criteria:**

Overall, the methods and evaluation criteria are well-chosen and appropriate for validating the paper's claims about improving LLM serving efficiency while maintaining performance. However, I have the following concerns:

1. The authors compare only against a single type of single-state baseline, which is quite simplistic. The gated-based attention models, for example, even though they are single-state could be capturing the historical context better. Ideally, the paper would be stronger if it was providing evidence that the dual state approach is useful for different types of recurrent attention mechanisms.

2. Studying the scaling effect to the performance of the proposed method is missing. There is no evidence that the approach works as the model size increases; the evaluation should demonstrate the relative effect even with limited budget (e.g. 1B, 4B, 7B, 12B).

3. Performing a qualitative analysis would be useful to better understand why there is degradation or improvement in certain datasets. It is somewhat surprising that the proposed method works better than full attention especially in code understanding and multi-doc QA.

**Other Comments Or Suggestions:**

See above

**Other Strengths And Weaknesses:**

See above.

**Questions For Authors:**

1. What is the training overhead cost of the proposed approach? Please mention how long does it take to train each layer and the total time it would take to convert the whole model.
2. Could the authors elaborate what is the intuition behind the dual attention state. Why it is sufficient and how layer conversion affects model capacity?
3. After which level of conversion does the quality of the model start to deteriorate?

**Relation To Broader Scientific Literature:**

C1: Dual-state linear attention

The literature in this area has focused mainly on single state mechanisms (e.g. Mamba, GLA). Using a dual state seems to be novel in this area and should be applicable to different types of methods in principle. Other than that this work makes use of well-established techniques such as GLA and theoretical (complexity) or empirical (recency bias) findings from prior work.

C2: On-the-fly layer conversion and efficient serving

There are offline methods focusing on reducing the complexity of attention through fine-tuning or layer-wise training (e.g. https://arxiv.org/abs/2103.13076). The on-the-fly conversion idea is interesting and is adaptive to different workloads but it is not compared to and positioned well with respect to other distillation methods (https://arxiv.org/abs/2305.02301), model compression methods (https://openreview.net/pdf?id=MxF0IKJtKW), or other inference optimization methods (flash attention, paged attention, KV caching).

**Theoretical Claims:**

The paper is empirical in nature and doesn't make any explicit theoretical claims. It only uses prior theoretical findings in the complexity analysis part. There are no formal proofs or theoretical arguments regarding the effect of conversion to model capacity and optimality of dual states.

---

> ### Author Rebuttal · Authors · 2025-04-01
>
> Thank you for the detailed and constructive comments! Please see below.
>
> **A1. Intuition behind dual state and model capacity**
> This work stems from our observation (Fig 1) that a single-state linear attention struggles to capture the full range of contextual information handled by self-attention. To address this, we introduce a dual-state mechanism that separately captures historical and recent context (Fig 5). Similar observations on foundation models’ memorization was also made in contemporary pre-training works (e.g. [6]). Our ablation (Tab 5) shows that more than two states yield limited gains. Despite the shift from self-attention, model capacity remains largely intact due to the dual states' expressiveness, as evidenced by strong results across LongBench (Tab 1), summarization (Tab 2), and Harness (Tab 3).
>
> **A2. Generalization: Application to a different linear architecture**
> Due to the space limit, please check A2 and Table C in our response to Reviewer a2iq.
>
> **A3. Scalability: Application to different model size**
> As requested, we evaluated our method on both 1.5B and 7B models. Results for the 7B model are in Tables 1–3 of the paper. Table A shows 1.5B results, where our distilled hybrid model outperforms the baseline Transformer (microsoft/phi-1_5) and 1.5B SSM model. Similar trends are observed at the 7B scale.
>
> Table A.
> | Model                | Token | Wino | ARC-e     | ARC-c     | PiQA      | HS |
> |---|---|---|---|---|---|---|
> | Transformer                 | 150B  | 73.4       | 75.0      | 48.0      | 76.6      | 62.6      |
> | SSM [1]                        | 3.0B  | 71.7       | 74.0      | 44.1      | 75.5      | 60.2      |
> | DSLA              | 800M  | **72.93** ± 0.01 | **74.49** ± 0.08 | **44.45** ±0.01 | **75.90**  ± 0.01 | **60.48** ± 0.0049 |
>
> **A4. On performance gain on Code understanding and Multi-doc QA**
> Tasks where DSLA performs better: Our method outperforms Zamba on code understanding, likely due to the strong coding understanding of our teacher model (LLaMA-2-7B). For Multi-doc QA, we even surpass the full-attention teacher—possibly because dual-state linear attention alleviates the "lost in the middle" issue. To support this, we ran a needle-in-a-haystack task and observed better retrieval when answers appeared between positions 1k–2k (e.g., DSLA retrieved 80% of answers at depth=19%, compared to 40% for the Transformer), Since Multi-doc QA often involves mid-context retrieval, DSLA is particularly effective.
>
> Tasks where DSLA performs worse: The performance drop may stem from approximation limitations in certain layers. We view deeper task-specific analysis as promising future work and will highlight these findings in the paper.
>
> **A5. Different deployment scenarios**
> While our method targets large-scale serving with multiple users, it also benefits single-user settings like PCs or edge devices. While the need for adaptiveness in large-scale systems arises from temporal variability and spatial imbalance, as explained in Sec. 3.2, the need for adaptiveness in edge devices stems from dynamic changes in power requirements [4], to meet a specific SLO. Static attention models may underperform under such variability, but DSLA adapts to changing hardware limits. To simulate an edge scenario, we ran a 340M model on a P100 GPU (12GB) and saw a 1.2× latency improvement over full attention when generating 1024 tokens from a 128 token prompt.
>
> **A6. Discussion on conversion rate**
> Importantly, our method highlights *progressive and selective conversion* to minimize accuracy degradation. Additionally, our framework allows setting a maximum conversion threshold to ensure accuracy is not significantly impacted.
>
> **A7. On performance deterioration**
> Empirically, we observed that performance begins to deteriorate when more than 75% of the model is converted, similar to [2]. However, this degradation can be mitigated through additional instruction tuning or Lora as in [3].
>
> **A8. Training overhead details**
> Training a single layer of a 7B model on 1B tokens takes ~5 hours on 4×A100 GPUs (80GB). Compared to full pretraining (858 days [5]), our fine-tuning costs just ~0.07%—a one-time expense.
>
> **A9. Statistical error information**
> We report the statistical error of End-to-end latency in Table B and lm harness result on Table A. We will update other experiments in final version.
>
> Table B.
> | Model     | Latency      |
> |---|---|
> | Llama2-7B | 93.64 ± 5.8  |
> | Zamba-7B  | 122.52 ± 8.6 |
> | DSLA-7B   | 40.83 ± 2.2  |
>
> [1] Transformers to ssms: Distilling quadratic knowledge to subquadratic models
>
> [2] The mamba in the llama: Distilling and accelerating hybrid models
>
> [3] LoLCATs: On Low-Rank Linearizing of Large Language Models
>
> [4] Dynamic-ofa: Runtime dnn architecture switching for performance scaling on heterogeneous embedded platforms
>
> [5] Llama: Open and efficient foundation language models
>
> [6] Leave no context behind: Efficient infinite context transformers with infini-attention

---

> > ### Comment · Reviewer_RVY3 · 2025-04-09
> >
> > Thank you for the replies. The rebuttal addressed most of my concerns: generalization to different architectures (via Mamba extension), varying model sizes (1.5B and 7B evaluations), insights about model improvement on code and long-doc tasks, statistical error reporting, and clarification on training overhead (minimal at ~0.07% of pretraining cost). For this reason, I decided to improve my score.

---

> > > ### Author Response · Authors · 2025-04-09
> > >
> > > Thank you so much for taking the time to review our rebuttal and for your thoughtful follow-up. We really appreciate your feedback and are glad to hear that our clarifications addressed your concerns.
> > >
> > > Best regards,
> > > Authors

---

### Official Review · Reviewer_9L4x · 2025-03-14

**Overall Recommendation:** 4

**Summary:**

This paper introduces Dual-State Linear Attention (DSLA), a novel attention mechanism designed to mitigate the short-range bias of traditional linear attention methods while maintaining the efficiency benefits necessary for long-context LLM serving. The key idea is to maintain two specialized hidden states, one for capturing historical context and another for tracking recency, allowing DSLA to balance global and local dependencies better than prior linear attention methods. To efficiently integrate DSLA into Transformer architectures, the paper proposes DSLA-Serve, an adaptive on-the-fly distillation framework that selectively replaces Transformer layers with DSLA layers during inference. This conversion is guided by sensitivity-based layer ordering, ensuring that DSLA layers are introduced progressively without degrading performance. The chained fine-tuning strategy ensures that each newly converted layer remains compatible with the previously replaced ones, preserving model consistency across partial conversions. Experiments demonstrate the effectiveness of DSLA.

**Claims And Evidence:**

Claim: Canonical Single-state linear attention overemphasizes recent tokens.
Evidence: observation in Fig. 1

Claim: DSLA enables model to flexibly trade off accuracy and efficiency.
Evidence: Tab. 1,2,3,4

**Essential References Not Discussed:**

No.

**Experimental Designs Or Analyses:**

I haven't spotted any problems with experiment designs. It uses a normal llm setting and compare with well-known baselines like GLA and Zamba.

**Methods And Evaluation Criteria:**

The normal benchmarks for LLMs are adopted.

**Other Comments Or Suggestions:**

The system overview in the appendix should be moved to the main paper for better understanding.

**Other Strengths And Weaknesses:**

Strengths:

1. Novelty: the method is novel compared with existing linear attention works.
2. Practical: since the method does not require re-training, it could be adapted to existing transformers.
3. Clearity: charts and plots demonstrate ideas clearly.

Weaknesses:

The authors are advised to add more baselines for comparison, including RetNet and Mamba.

**Questions For Authors:**

Apart from latency, is it okay to report and compare FLOPs?

**Relation To Broader Scientific Literature:**

In contrast to previous one-stage linaer attention, this paper proposes a two-stage solution that is flexible for use and efficient.

**Theoretical Claims:**

Not applicable.

---

> ### Author Rebuttal · Authors · 2025-04-01
>
> Thank you for the detailed and constructive comments! Please see below.
>
> **A1. Comparison with RetNet and Mamba**
> We added more baselines including RetNet [1] and Mamba [2], and measured zero-shot performance on challenging tasks including PiQA, ARC-challenge, Hellaswag (HS), MMLU, Winogrande.  On average, DSLA (ours) outperforms both RetNet and Mamba.
>
> Table A.
> | Model | PiQA | ARC | HS | MMLU | WG | Avg |
> |:---:|:---:|:---:|:---:|:---:|:---:|:---:|
> | RetNet 6.7B | 77.8 | 39.9 | 72.9 | 26.1 | 66.1 | 56.56 |
> | Mamba 7B | 78.3 | 42.8 | **77.8** | 33.0 | **71.9** | 60.76 |
> | Ours | **78.7** | **43.2** | 75.4 | **46.1** | 69.9 | **62.66** |
>
> **A2. FLOPs Report**
> Thanks for the great suggestions. Please note that we primarily reported end-to-end latency in our paper, as FLOPs reflect only the total computation per inference and do not account for dynamic load, memory bandwidth, or parallelism. Following table shows FLOPs measured with  torchprofile [3] by feeding a fixed input of length 12 and performing a single token generation. Since FLOPs scale linearly with the number of activated parameters, Zamba-7B exhibits higher FLOPs due to its repeated activation of shared attention layers.
>
> Table B.
> | Model | Architecture | GFLOPS |
> |---|---|---|
> | Llama-2-7B | Transformer | 158.6 |
> | Mamba 7B | SSM | 125.6 |
> | Zamba 7B | Hybrid | 277.7 |
> | DSLA 7B (100% converted) | Dual-state Linear Attention | 163.5 |
>
> We will reorganize the figure’s location on our revised version. Thanks for your thoughtful comments!
>
>
> **Reference**
>
> [1] Sun, Yutao, et al. "Retentive network: A successor to transformer for large language models." arXiv preprint arXiv:2307.08621 (2023).
>
> [2] Gu, Albert, and Tri Dao. "Mamba: Linear-time sequence modeling with selective state spaces." arXiv preprint arXiv:2312.00752 (2023).
>
> [3] https://github.com/zhijian-liu/torchprofile

---

### Decision · Program_Chairs · 2025-05-01

**Decision:**

Accept (poster)

**Comment:**

The paper presents a novel and practically effective approach to improving long-context LLM serving efficiency.
While baseline comparisons and scalability analysis are limited, the empirical results and methodological contributions are strong.
I would like to recommend conditional acceptance pending the following revisions:
1. add comparisons with RetNet, Mamba, and other recent hybrid architectures.
2. Clarify training overhead (e.g., time/resource costs for fine-tuning DSLA layers).
3. Discuss scalability limitations and potential solutions (e.g., GPU memory constraints).